# Health-Related Quality of Life after Adolescent Fractures of the Femoral Shaft Stabilized by a Lateral Entry Femoral Nail

**DOI:** 10.3390/children9030327

**Published:** 2022-03-01

**Authors:** Thoralf Randolph Liebs, Anna Meßling, Milan Milosevic, Steffen Michael Berger, Kai Ziebarth

**Affiliations:** Inselspital, Department of Paediatric Surgery, University of Bern, 3010 Bern, Switzerland; geppert.anna@gmail.com (A.M.); milan.milosevic@insel.ch (M.M.); steffen.berger@insel.ch (S.M.B.); kai.ziebarth@insel.ch (K.Z.)

**Keywords:** fracture, femur, surgery, nail, intramedullary, health-related quality of life

## Abstract

(1) Background: In adolescents, fractures of the femoral shaft that are not suitable for elastic-stable-intramedullary-nailing (ESIN), are challenging. We aimed to evaluate the health-related quality of life (HRQoL) and complications in adolescents treated with intramedullary rodding using the adolescent lateral trochanteric entry femoral nail (ALFN), and to assess if HRQoL was associated with additional injuries. (2) Methods: We followed-up on 15 adolescents with a diaphyseal femoral fracture who were treated with an ALFN from 2004 to 2017. Patients were asked to fill in a questionnaire that includes the iHOT, Peds-QL, and the Pedi-IKDC. (3) Results: The ALFN was used as a primary method of fixation in 13 patients, and as a fixation for failed ESIN in two cases. All 15 fractures healed radiographically. One distal locking screw broke. After a mean follow-up of 2.8 years, the mean iHOT-12 was 14.0 (SD 15.4), PedsQL-function was 85.7 (SD 19.3), PedsQL-social-score was 86.2 (SD 12.5), and the mean Pedi-IKDC was 77.2 (SD 11.3). In patients where the femoral fracture was an isolated injury, the HRQoL-scores were consistently higher compared with patients who sustained additional injures. (4) Conclusions: Treating diaphyseal fractures in adolescents with an ALFN resulted in good radiographic outcomes in all our cases. HRQoL, as measured by the iHOT, PedsQL, and Pedi-IKDC, was good to excellent; but it was consistently inferior in patients with additional injuries. These results suggest that the ALFN is a good alternative when patients are not suitable for ESIN, and that the HRQoL of adolescents who were treated with an ALFN is mainly influenced by the presence of additional injures, and less by the fracture of the femur itself.

## 1. Introduction

Since fractures of the femoral shaft are not too common among children (e.g., 0.89% [1]), many paediatricians and orthopaedic surgeons may have only limited experience in treating these injuries. Nonetheless, given the need for emergency surgery in children 3 years of age and older, every orthopaedic surgeon may be confronted with the need to perform surgery in these children.

The preferred treatment strategy for paediatric femoral fractures is age dependent, ranging from bandage immobilization in new-borns to overhead extension or hip spica cast in children up to about 3 years of age. For older children up to the teenage years, elastic intramedullary nailing (ESIN) is the standard treatment. However, in adolescents the management of diaphyseal femoral fractures is highly debated. Treatment options in that age group include ESIN, external fixation [2], a combination of these two methods, submuscular plating [3], or rigid intramedullary rodding.

Several studies have demonstrated that, due to its flexible nature, elastic intramedullary nailing has an increased risk of complications in heavier or older children. Commonly, a body weight of 50 kg [4,5] or 55 kg [6] is considered as a threshold in that respect. On the other hand, plate fixation involves incisions that might be regarded as unattractive, and plate fixation has been associated with valgus deformity, especially if the plate is not removed [3].

Intramedullary rodding through the piriform fossa, a standard approach known to every orthopaedic surgeon, has been associated with reports of avascular necrosis of the femoral head [7,8,9] which ultimately may lead to total hip arthroplasty. For this reason, this technique has not been recommended in adolescents.

With advanced knowledge of the blood supply of the femoral head, entry points lateral to the tip of the greater trochanter have been evaluated. These entry points have been reported not to be associated with avascular necrosis. Given these promising reports, Gordon et al. used a rigid intramedullary nail that was designed for fractures of the humerus, and used this humeral nail for the treatment of femoral shaft fractures in 15 children and adolescents [10]. He reported that this technique was “safe, effective and well-tolerated” [10]. Keeler et al. used a rigid interlocking paediatric femoral nail that was based on the design of the modified humeral interlocking nail and which was introduced through the lateral aspect of the greater trochanter. These authors followed 24 fractures and looked at radiographic outcomes. They did not report any avascular necrosis of the femoral head as well [11].

Both these devices were not specifically designed for the treatment of adolescent femoral fractures. Led by the experience gained in adults with helically shaped intramedullary nails, a titanium cannulated adolescent lateral entry femoral nail (ALFN) was developed. Reynolds et al. used this device in 15 patients and reported both a shorter recovery time for patients treated with ALFN in comparison to elastic nailing and a low rate of complications, without major complications [12].

Although these results are promising, rigid nailing with a specially designed paediatric lateral trochanteric entry femoral nail does not appear to be generally accepted for this patient population currently. For example, in the most current guideline of the American Academy of Orthopaedic Surgeons (AAOS) as of December 2020 it was written: “There is currently insufficient literature in specially designed paediatric rigid intramedullary nails […] for inclusion in the current guideline…Limited evidence supports rigid trochanteric entry nailing, submuscular plating, and flexible intramedullary nailing as treatment options for children age eleven years to skeletal maturity diagnosed with diaphyseal femur fractures, but piriformis or near piriformis entry rigid nailing are not treatment options” [13]. In a recent retrospective multicentre study of 16 centres in Germany, in which 53 children with femoral fractures were analysed, only three patients were treated with a lateral trochanteric entry femoral nail [14]. The majority of the other patients were treated with ESIN, of whom eight were revised. Other reported treatments in that report were primary of secondary plates (nine cases), intramedullary nails of adult traumatology, or external fixators [14].

As an additional aspect, none of the studies mentioned above assessed the health-related quality of life (HRQoL) in these patients up to this time.

Therefore, we initiated this study to assess the treatment results of patients that were treated with an ALFN in terms of HRQoL, radiographic healing, and complications. In addition, we aimed to evaluate if HRQoL was associated with additional injuries.

## 2. Materials and Methods

This is a retrospective analysis of clinical and radiographic results, in which patients, who underwent treatment for a diaphyseal fracture of the femur by an ALFN, were contacted by postal mail.

Several methodological details are identical to sister studies in which the health-related quality of life (HRQoL) after fractures of the lateral third of the clavicle, proximal humerus or supracondylar humerus in children and adolescents was assessed [15,16,17].

All sequential patients up to 16 years of age, who were treated at our institution with an ALFN for a diaphyseal fracture of the femur during the period January 2004 to April 2017 were candidates for inclusion in the study. Our institution is one of the leading paediatric trauma centres in the Switzerland, serving more than one million inhabitants.

Patients were identified based on the radiological reports within our Picture Archiving and Communication System (PACS).

For the purpose of this analysis, the inclusion criteria were limited to patients who have sustained a diaphyseal femur fracture. Exclusion criterion was the inability to complete the questionnaires because of cognitive or language difficulties (Figure 1). We did not exclude patients with additional injuries.

A radiological analysis was performed to classify the fracture according to the following criteria: AO classification scheme [18], length stability using the Winquist–Hansen classification system [19], open physis of the greater trochanter, fracture healing, avascular necrosis of the femoral head, and growth disturbance of the greater trochanter. The physicians performing the image analysis were not aware of the patient’s clinical result, thereby avoiding observer bias.

Beginning in 2016 we sent information about the study, a consent form, and questionnaires to the patients by postal mail (Figure 1). Non-responding participants were reminded three times by mail. Participants still not responding were contacted by phone to determine the reason for non-responding. At that time, it was attempted to administer the questionnaire by phone.

As there are currently no outcome instruments described in the literature to specifically assess the HRQoL after fractures of the femur, we used disease specific outcome measures of the hip and knee instead. For assessing outcomes after injuries around the hip, we chose the International Hip Outcome Tool 12 (iHot-12) [20], which is available in a validated translated version in German. That outcome was reported to provide good validity, reliability, and responsiveness for the evaluation of physically active patients with a hip disorder [20]. There are several other outcome measures regarding hip diseases available; however, none of these were validated in the paediatric population [21]. For assessing the outcome of fractures around the knee, we chose the Pedi-IKDC [22,23], which was reported to have better psychometric properties than the KOOS-Child [24].

As secondary outcomes, we selected the non-disease specific Paediatric Quality of Life Inventory (PedsQL) [25] which is available in a validated translated version. Scores were standardised to 0–100, with higher scores indicating more physical or more social function.

Data on demographics, dates of the injury, the side (right/left), mechanism of the injury, and the treatment course were collected from both the radiological analysis and from the electronic patient chart. In the questionnaire, we included items about concomitant injuries.

Closed reduction and fixation with the ALFN was performed according to the manufacturer’s instructions (Synthes, Oberdorf, Switzerland). Special attention was provided to avoid the piriform fossa as an entry point and to use the lateral aspect of the greater trochanter as the entry point instead, typically at the same level as projected to the cranial border of the base of the femoral neck. When selecting the entry point it is important to consider the diameter of the drill (13.0 mm) that will be used for reaming to avoid injury to the femoral circumflex vessels. Postoperatively, patients were allowed weight-bearing as tolerated using crutches. Physiotherapy was used in every patient for instructions on mobilization and for assuring a good range of motion of the ipsilateral hip and knee.

All patients were invited to a routine consultation visit after 4 weeks. At that time, the patients typically were able to walk without crutches and demonstrated a good range of motion of the hip and knee. We usually recommended removal of the implant after 1 year.

After the description of the main outcome measure, we performed a bivariate analysis in which we analysed the HRQoL and other factors in relation to the presence of additional injuries. We used the non-parametric Mann–Whitney U Test for comparisons. All *p*-values are two-tailed; no corrections were made for multiple comparisons. A statistical analysis was performed using SPSS (SPSS Inc., Chicago, IL, USA).

## 3. Results

We were able to follow-up on 15 patients (6 girls, 9 boys) who were treated with an ALFN at an average age of 14.0 (SD 1.0) years of age. The mean body weight was 55 kg (SD 7, range from 40 to 68 kg) and the mean body height was 165 cm (SD 8 cm) (Table 1). The ALFN was used as a primary method of fixation in 13 patients, and as a fixation for failed ESIN in two cases (Figure 2 and Figure 3). All 15 fractures healed radiographically. Physis of the greater trochanter were open in eight cases. There were no avascular necrosis of the femoral head and no growth disturbances of the greater trochanter. Complications consisted of a broken distal locking screw in one case, of which a fragment remained in situ during implant removal (Figure 4).

After a mean follow-up of 2.8 years (SD 2.6, range 0.3 to 7.2 years), the mean iHOT-12 score was 14.0 (SD 15.4), at a scale of 0–100, with lower values representing better HRQoL. The mean function score of the PedsQL was 85.7 (SD 19.3), and the mean social score of the PedsQL was 86.2 (SD 12.5), both at a scale of 0–100, with higher values representing better HRQoL. The mean Pedi-IKDC was 77.2 (SD 11.3) (Table 2).

In patients in whom the femoral fracture was an isolated injury, the HRQoL-scores were consistently higher when compared with patients who sustained additional injures (Table 2). However, provided the low number of patients, that difference was not statistically significant.

## 4. Discussion

The treatment of fractures of the femoral shaft in adolescents can be challenging. Especially when the intramedullary canal is narrow, it is often not possible to insert elastic intramedullary nails that are strong enough to withstand the forces that act on the femur in teenagers weighing 50 kg or more. This study showed that an adolescent lateral trochanteric entry femoral nail can be used in these cases, for both primary fixation and revision of failed elastic intramedullary fixation, as it led to radiological consolidation of all 15 patients that were analysed in this study. In addition, this study demonstrated that these patients have good health-related quality of life (HRQoL) as measured with the iHOT-12, the IKDC, and the Peds-QL at a mean follow-up of 2.8 years.

Despite these advantages, specifically designed adolescent lateral trochanteric femoral nails, such as den ALFN, are not yet the standard care in all institutions dealing with femoral fractures in adolescents [13,14].

### 4.1. Limitations

There are several limitations to this study: First, this was a mono-centre study, suggesting limited external validity. However, we are the only hospital treating paediatric trauma in a greater geographical area and all sequential patients were included in our study. This should reduce the probability of a bias in the run-in phase, making a high external validity probable [15]. In addition it has been stated that no single study is capable of providing full external validity, since it has been reported that great variation exists across and within countries for orthopaedic treatments [26]. Second, we had only 15 cases. This might be considered a small case series. However, all other publications on the HRQoL after femoral fractures in this age group treated with intramedullary rodding did not report on more patients than we reported. Third, the examined radiographs were not specifically prepared for this analysis, but were made routinely. Therefore the quality of these radiographs is comparable to the situation of the clinician [15]. Moreover, the physician classifying the fractures was not aware of the clinical result of the patient, thereby the radiological assessment could be regarded as blinded [15]. Fourth, as this study has a retrospective design it suffers from typical methodological weaknesses, such as no intermediate data points and missing data on the HRQoL prior to the injury. While the latter is considered a methodological weakness in studies analysing adult fractures, this does not necessarily apply to adolescent fractures, as adolescents usually have no physical limitations before the injury. Therefore it can be assumed that limitations of the disease-specific outcome measure are in fact attributable to the injury [15]. Fifth, there is no disease-specific outcome instrument for assessing the HRQoL after fractures of the femur. Therefore, we used accepted disease-specific outcome instruments for the adjacent hip and knee joint. Unfortunately, this limits the number of comparable literature considerably. Sixth, our follow-up rate was 83%. This rate is above the recommended 80% that is commonly used as a threshold and most other studies we are aware of have a lower rate of follow-up, if the follow-up is reported at all. In addition, we were not able to identify another study that assessed the HRQoL in a comparable patient group. Seventh, we do not know how the ALFN compares to other treatment options in terms of HRQoL. As we do not have a comparison group in our study and we are not aware of comparable publications in the literature, that question will be subject of further studies. Eighth, we did not exclude patients with additional injuries, as we wanted to report the results of this procedure on all subsequent patients in whom we used this device. Therefore, we stratified the results according to additional injures in the tables so that readers who are interested in the results of patients who did not sustain additional injuries can be referred to the tables.

### 4.2. Demographics, Radiographic Analysis and Complications

As can be seen in Table 1, the average patient who was treated with an ALFN in this study was 14 years old of age, had a body height of 165 cm, and had a body weight of 55 kg. These demographics are comparable to the literature [27]. All fractures healed radiographically and there was no evidence of avascular necrosis of the femoral head or growth disturbances of the greater trochanter. These results are compatible with Keeler [11], who also did not report such complications in her innovative study when she used humeral nails for femoral fractures. Apart from one screw breakage during implant removal, which can be attributed to the fact that only one instead of two locking screws was used, there were no complications. A fragment of that screw remained in the intramedullary canal (Figure 4).

### 4.3. Health-Related Quality of Life and AO Classification

The HRQoL in this study as assessed by the iHOT-12, the IKDC, and the Peds-QL was good, but not perfect. This was true for both the disease-specific and the non-disease-specific outcome measures. This indicates that there could be a limitation in HRQoL after these injuries. This is compatible with the literature, as other authors have also described that there may be significant reduction in HRQoL after fractures of the lower limb. For example, in a study with 162 children, of which 54.8% had femoral fractures, there were physical function scores that were lower than age-matched norms at 6 months after the injury [28].

### 4.4. HRQoL and Other Injuries

The HRQoL scores were consistently better in the group without additional injuries compared with the adolescents who sustained additional injuries. This indicates that the HRQoL of adolescents who were treated with an ALFN is mainly influenced by the presence of additional injures, and less by the fracture of the femur itself.

## 5. Conclusions

The ALFN is a feasible treatment option in the adolescent population for the treatment of femoral shaft fractures, especially when patients are overweight or have a narrow intramedullary canal. Our study showed excellent health-related quality of life and low rates of adverse events, there were no cases of AVN or disturbance of trochanteric growth. Given further advantages of the ALFN, such as less soft tissue injury compared with plate fixation, the possibility of immediate full weight-bearing, low risk of non-union, and the avoidance of an external fixator, we prefer the ALFN for the treatment of femoral shaft fractures in cases when ESIN is not suitable.

## Figures and Tables

**Figure 1 children-09-00327-f001:**
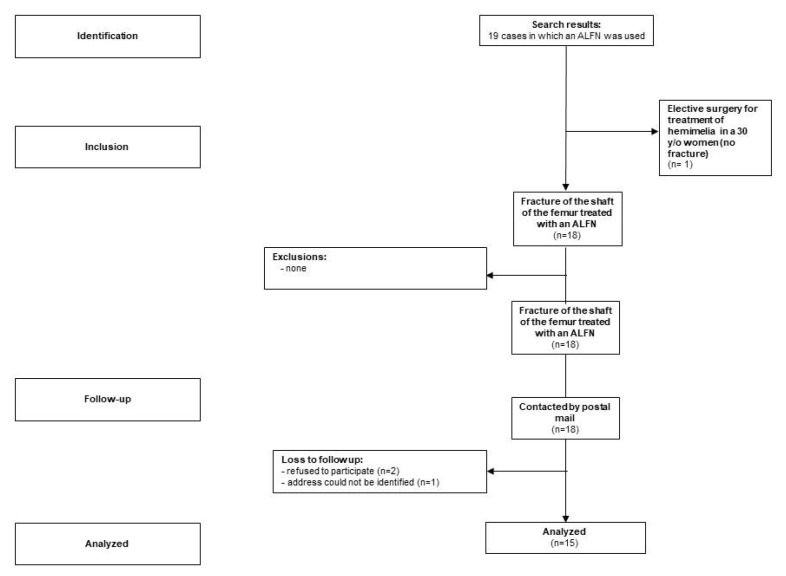
STROBE Participant flow chart.

**Figure 2 children-09-00327-f002:**
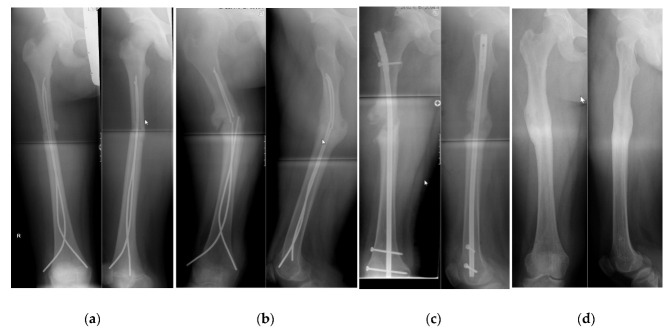
(**a**) 1 week after ESIN for a femoral fracture. Please note the narrow intramedullary canal, prohibiting the use of nails with a thicker diameter. Although currently we would advance the elastic intramedullary nail further, we do not consider this a reason for the subsequent failure of this fixation. (**b**) Same patient as in Figure 2a, now 9 months after ESIN. There is failure of the elastic nail due to non-union, resulting in a malposition. (**c**) Now 1 week after revision with an ALFN. Note the correction of the malposition. (**d**) After removal of the ALFN: radiographic bony union in correct position.

**Figure 3 children-09-00327-f003:**
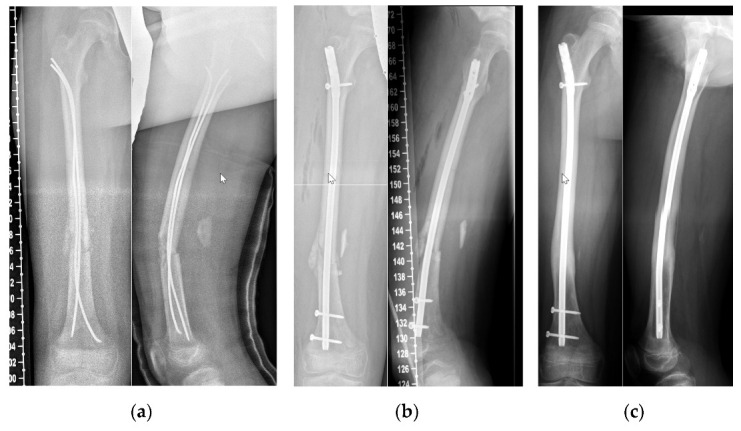
(**a**) Three days after anterograde ESIN for a femoral fracture. Due to additional injuries an anterograde approach was used. For this reason, it was not possible to achieve as much tension at the level of the fracture as we would have desired. (**b**) Same patient as in Figure 3a, six weeks after the anterograde ESIN fixation we revised to an ALFN. (**c**) Now eight months after revision to ALFN, with union in correct position.

**Figure 4 children-09-00327-f004:**
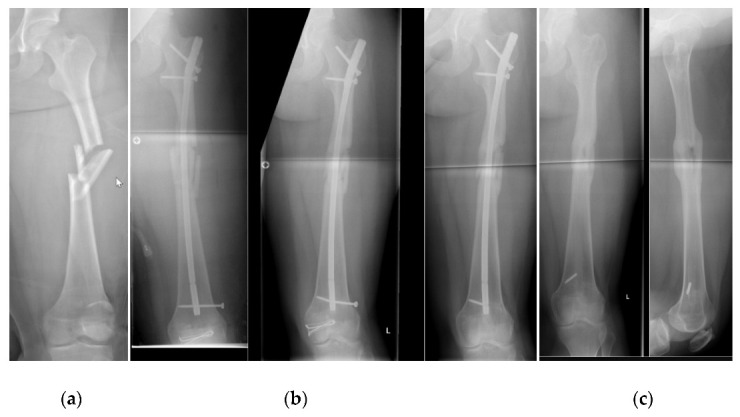
(**a**) Initial radiograph of a segmental fracture of the femoral shaft without cortical contact. (**b**) Same patient as in Figure 4a, four weeks after fixation with an ALFN. The position of the femoral neck screw was tolerated as it did not disturb the patient clinically. Only one distal locking screw was used which could have been advanced further. (**c**) Now after removal of the ALFN. Note the medial fragment of the distal locking screw, which was tolerated.

**Table 1 children-09-00327-t001:** Baseline characteristics by additional injury.

		Additional Injury																	
		No Additional Injury						With Additional Injury						Total					
		*n*	Column N %	Mean	SD	Min	Max	*n*	Column N %	Mean	SD	Min	Max	*n*	Column N %	Mean	SD	Min	Max
Gender																			
	female	3	33%					3	50%					6	40%				
	male	6	67%					3	50%					9	60%				
Age at the time of injury [years]		9		13.8	1.0	12.5	15.6	6		14.2	1.0	13.1	15.9	15		14.0	1.0	12.5	15.9
Weight [kg] at time of the injury		9		52	6	40	60	6		59	6	50	68	15		55	7	40	68
Height [cm] at time of the injury		9		164	10	153	178	6		166	5	160	171	15		165	8	153	178
BMI [kg/m^2^] at time of the injury		9		19.1	2.0	17.1	23.3	6		21.7	2.2	19.5	25.9	15		20.2	2.4	17.1	25.9
Injured side (right vs. left)																			
	right	4	44%					3	50%					7	47%				
	left	5	56%					2	33%					7	47%				
	bilateral							1	17%					1	7%				
Radiological classification according to the AO																			
	32-D/4.1	3	33%					1	17%					4	27%				
	32-D/5.1	5	56%					3	50%					8	53%				
	32-D/5.2	1	11%					2	33%					3	20%				
Winquist and Hansen classification regarding the degree of comminution																			
	0: Transverse or short oblique fractures with no comminution	4	44%					1	17%					5	33%				
	1: Small butterfly fragment of less than 25% of width of the bone	3	33%					2	33%					5	33%				
	2: Butterfly fragment of 50% or less of the width of the bone							2	33%					2	13%				
	3: Large butterfly fragment greater than 50% of the width of bone	1	11%											1	7%				
	4: Segmental comminution	1	11%					1	17%					2	13%				
ALFN as the primary treatment or as a revision																			
	ALFN used for revision of otherwise failed fixation	1	11%					1	17%					2	13%				
	ALFN used as primary fixation	8	89%					5	83%					13	87%				
Injury mechanism																			
	motor vehicle accident	2	22%					6	100%					8	53%				
	sports	4	44%											4	27%				
	fall from tree/play	3	33%											3	20%				
Was the skin injured at the time of the injury?																			
	No, skin was intact	9	100%					5	83%					14	93%				
	Yes, but was just a scratch																		
	Yes, a suture was necessary							1	17%					1	7%				

**Table 2 children-09-00327-t002:** Follow-up data by additional injury.

		Additional Injury
		No Additional Injury	With Additional Injury	Total
		Mean	SD	Min	Max	Count	Column N %	Mean	SD	Min	Max	Count	Column N %	Mean	SD	Min	Max	Count	Column N %
Follow-up duration [years]	2.94	2.91	0.45	7.16	9		2.55	2.29	0.29	5.63	6		2.79	2.6	0.29	7.16	15	
iHOT-12 (0–100)	12.9	14.8			9		15.7	17.4			6		14	15.4			15	
IKDC	80.2	7.54			9		73.1	14.8			6		77.2	11.3			15	
PedsQL physical function	91.8	9.02			9		76.6	27.2			6		85.7	19.3			15	
PedsQL social function	90.8	7.72			9		79.2	15.7			6		86.2	12.5			15	
Are you satisfied with the thigh that was injured?													
	Very satisfied					5	56%					4	67%					9	60%
	A little satisfied					3	33%					2	33%					5	33%
	A little unsatisfied					1	11%											1	7%
	Very unsatisfied																		
Are you satisfied with the treatment that was performed?													
	Very satisfied					5	71%					3	75%					8	73%
	A little satisfied					2	29%					1	25%					3	27%
	A little unsatisfied																		
	Very unsatisfied																		
Now you know the treatment and the results. If you could turn back time, would you choose this treatment again?						
	Yes, definitely					5	71%					3	75%					8	73%
	Yes, probably					2	29%					1	25%					3	27%
	No, probably not																		
	No, not at all																		
How would you describe the pain that you typically experience in your thigh?										
	No pain					2	22%					5	83%					7	47%
	Little pain					6	67%											6	40%
	Moderate pain					1	11%					1	17%					2	13%
	Strong pain																		
When does the pain typically occur?																
	I do not have any pain				4	44%					4	67%					8	53%
	Only for the first steps				1	11%					1	17%					2	13%
	Only after longer walks (30 min)			4	44%					1	17%					5	33%
	When walking																		
	Constant pain

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
