# Peer review of "Health-Related Quality of Life after Adolescent Fractures of the Femoral Shaft Stabilized by a Lateral Entry Femoral Nail"

_children, 2022, doi:10.3390/children9030327_

Round 1
Reviewer 1 Report
The subject is interesting for everyone treating fractures in adolescence. Unfortunately the reader can't integrate the data in comparable data from healthy adolecents or patients with ESIN or with any other fracture (ankle, knee, hip). With your responsibility for a wide geographical area you will have patient groups to give comparable data.
line 45: MIPO prevents rather long incisions
line 46: overgrowth in adolescents?
line 58: If you mention "Gordon" by name, you should mention "another author" by name in the same way, it was Kathryn Keeler
line 61: SHE did not report ...
line 78-81: 53 children - 3 ALFN, 8 rigid nail, plates 9, EF 11 - 22 other cases??
line 83: as an additional problem? aspect sounds better
line 150-152: these informations are results
line 189 ff.: Fig. 3 gives no more information than Fig. 2 (dispensable)
Author Response
Reviewer: The subject is interesting for everyone treating fractures in adolescence.
Authors responsonse:
Thank you very much!!!
Reviewer:
Unfortunately the reader can't integrate the data in comparable data from healthy adolecents or patients with ESIN or with any other fracture (ankle, knee, hip). With your responsibility for a wide geographical area you will have patient groups to give comparable data.
Authors response:
This is a very important point. As to now, we have comparable data regarding the Peds-QL of fractures of the lateral third of the clavicle, proximal humerus, and supracondylar humerus. The respective values are listed in the table below:
|
Fracture site |
subgroup |
Peds-QL PF |
Peds-QL PS |
|
|
Current Study |
femur, shaft |
ALFN without further injuries |
91,8 |
90,8 |
|
Current Study |
femur, shaft |
ALFN with additional injuries |
76,6 |
79,2 |
|
Liebs, 2019 |
clavicle, lateral third |
pts aged 12 yrs and older |
94,8 |
88,0 |
|
Liebs, 2021 |
humerus, proximal |
pts aged 12 yrs and older |
93,6 |
90,9 |
|
Liebs, 2020 |
humerus, supracondylar |
all ages (mean 6.1 years) |
97,2 |
93,3 |
As can be seen in this table, the values of the Peds-QL psychosocial function (PS) are quite comparable when looking at the patients who did NOT have additional injures other than the fracture of the femoral shaft. A similar scenario is seen when looking at the physical function scale (PF). Please note, that the best values for both scores are seen in the patients who sustained a supracondylar fracture of the humerus. However, these children were much younger at the time of the injury (mean 6.1 years).
Reviewer: line 45: MIPO prevents rather long incisions
Authors response: We have deleted „rather long“.
Reviewer: line 46: overgrowth in adolescents?
Authors response: The paper in reference 3 states: „bony overgrowth of the plate“
Reviewer: line 58: If you mention "Gordon" by name, you should mention "another author" by name in the same way, it was Kathryn Keeler
Authors response: Thank you very much. We have added her name now.
Reviewer: line 61: SHE did not report ...
Authors response: Thank you very much. We have corrected it, of course.
Reviewer: line 78-81: 53 children - 3 ALFN, 8 rigid nail, plates 9, EF 11 - 22 other cases??
Authors response: Very good point. Unfortunately the numbers in the report do not easily add up. Most other cases received ESIN, mostly with two, some with three nails. We have changed the text accordingly.
Reviewer: line 83: as an additional problem? aspect sounds better
Authors response: Thank you, we have changed it.
Reviewer: line 150-152: these informations are results
Reviewer: line 189 ff.: Fig. 3 gives no more information than Fig. 2 (dispensable)
Authors response: As per the reviewers suggestion, we have now removed Fig. 3 and changed the numbering of Fig. 4 to Fig. 3.
We would like to thank reviewer 1 for the knowledgeable and thoughtful comments!
Reviewer 2 Report
The paper explores the usage of lateral entry femoral nail in a population of pre-adolescent and adolescent femur shaft fractures.
The authors delivered a very well written report of their experience and I really appreciate the inclusion of HRQoL for assessing the outcome; I think that this kind of questionnaires are very useful in pediatric patients.
I have just some minor revision to ask:
Line 46-47: The authors state that "plate fixation has been associated with overgrowth of the femur leading to leg length discrepancies, especially if the plate is not removed" and enclose a citation to support it. However, the paper cited do not state such a thing.
Kelly et al. addressed in particular valgus deformity and stress shielding for plate retention. They reported limb lenght discrepancy as a generic complication but they did not blame the plate.
Limb lenght discrepancy is, in fact, a very common consequence of femur (or tibia) fracture in children despite the treatment used.
I suggest to modify the lines with "and has been associated with valgus deformity, especially if the plate is not removed"
Line 113 and line 245: I suggest to change "persons" and "person" with "physicians" and "physician".
Line 240-242: The authors correctly recognize the small number of children in the case series as a limitation but they add that "all other publications on this subject did not report on more patients than we did".
That's not true. In particular, they correctly cite the work of Keeler et al. that reported 80 fractures afterwards in the discussion. Furthermore, also Elgohary et al. (Elgohary HS, El Adl WA. Antegrade rigid nailing through the tip of the greater trochanter for pediatric femoral shaft fractures. Eur J Orthop Surg Traumatol. 2014 Oct;24(7):1229-35. doi: 10.1007/s00590-013-1382-z. Epub 2013 Dec 4. PMID: 24306168.) reported 23 children treated the same way. I suggest to remove that assertment.
Finally, the authors state that no growth disturbances of the greater trochanter in the series.
I'd like to know how many patients had, at the time of surgery, open physis of the great trochanter (in the x-ray showed, it's not clear) and I think that adding this data could be useful maybe in the material and methods chapter.
Author Response
Reviewer:
The paper explores the usage of lateral entry femoral nail in a population of pre-adolescent and adolescent femur shaft fractures.
The authors delivered a very well written report of their experience and I really appreciate the inclusion of HRQoL for assessing the outcome; I think that this kind of questionnaires are very useful in pediatric patients.
Authors response:
Thank you very much!!!
Reviewer:
I have just some minor revision to ask:
Line 46-47: The authors state that "plate fixation has been associated with overgrowth of the femur leading to leg length discrepancies, especially if the plate is not removed" and enclose a citation to support it. However, the paper cited do not state such a thing.
Kelly et al. addressed in particular valgus deformity and stress shielding for plate retention. They reported limb lenght discrepancy as a generic complication but they did not blame the plate.
Limb length discrepancy is, in fact, a very common consequence of femur (or tibia) fracture in children despite the treatment used.
I suggest to modify the lines with "and has been associated with valgus deformity, especially if the plate is not removed"
Authors response:
Thank you very much for that point. We have changed the sentence accordingly.
Reviewer:
Line 113 and line 245: I suggest to change "persons" and "person" with "physicians" and "physician".
Authors response:
Thank you very much. We have changed it.
Reviewer:
Line 240-242: The authors correctly recognize the small number of children in the case series as a limitation but they add that "all other publications on this subject did not report on more patients than we did".
That's not true. In particular, they correctly cite the work of Keeler et al. that reported 80 fractures afterwards in the discussion. Furthermore, also Elgohary et al. (Elgohary HS, El Adl WA. Antegrade rigid nailing through the tip of the greater trochanter for pediatric femoral shaft fractures. Eur J Orthop Surg Traumatol. 2014 Oct;24(7):1229-35. doi: 10.1007/s00590-013-1382-z. Epub 2013 Dec 4. PMID: 24306168.) reported 23 children treated the same way. I suggest to remove that assertment.
Authors response:
Thank you very much for this remark. We have now changed the sentence as follows:
„However, all other publications on the HRQoL after femoral fractures treated with intramedullary rodding in this age group did not report on more patients than we did.“
We hope that this wording is accetable to the reviewer.
Reviewer:
Finally, the authors state that no growth disturbances of the greater trochanter in the series.
I'd like to know how many patients had, at the time of surgery, open physis of the great trochanter (in the x-ray showed, it's not clear) and I think that adding this data could be useful maybe in the material and methods chapter.
Authors response:
Following the reviewers suggestion, we have re-analyzed our radiographs: There we found that the physis of the greater trochanter was open in eight cases. We have added this information in the results section.
We would like to thank reviewer 2 for the thoughtful and knowledgeable comments!
Round 2
Reviewer 1 Report
line 62 ff.: Keeler et al.; the authors; they did not report...
line 85 ff.: the numbers of the mentioned study are still not correct resp. can't be understand: 3 ALFN + 31 ESIN + ......
Author Response
Reviewer:
line 62 ff.: Keeler et al.; the authors; they did not report...
Authors responsonse: Following the reviewers suppestion, we have changed that passage.
Reviewer:
line 85 ff.: the numbers of the mentioned study are still not correct resp. can't be understand: 3 ALFN + 31 ESIN + ......
Authors responsonse:
The reviewer is correct: the numbers do not add up. Unfortunately, the numbers do not add up in the original report that we have referenced here.
For this reason we suggest to rephrase that passage as follows:
… And in a recent retrospective multicentre study of 16 centres in Germany, in which 53 children with femoral fractures were analysed, only three patients were treated with a lateral trochanteric entry femoral nail.[14] The majority of the other patients were treated with ESIN, of whom eight had to revised. Other treatments in that report were primary of secondary plates (nine cases), intramedullary nails of adult traumatology, or external fixators.[14]
We would like to thank reviewer for the good comments again! These helped us to improve the manuscript considerably.